# Medication Review: What’s in a Name and What Is It about?

**DOI:** 10.3390/pharmacy12010039

**Published:** 2024-02-19

**Authors:** Anneleen Robberechts, Maja Brumer, Victoria Garcia-Cardenas, Niurka M. Dupotey, Stephane Steurbaut, Guido R. Y. De Meyer, Hans De Loof

**Affiliations:** 1Laboratory of Physiopharmacology, University of Antwerp, Universiteitsplein 1, 2610 Antwerp, Belgium; maja.brumer@uantwerpen.be (M.B.); guido.demeyer@uantwerpen.be (G.R.Y.D.M.); hans.deloof@uantwerpen.be (H.D.L.); 2Meduplace, Royal Pharmacists Association of Antwerp (KAVA), 2018 Antwerp, Belgium; 3Centre for Pharmaceutical Research, Research Group of Clinical Pharmacology and Clinical Pharmacy, Vrije Universiteit Brussel, Laarbeeklaan 103, 1090 Jette, Belgium; stephane.steurbaut@uzbrussel.be; 4Pharmacy and Pharmaceutical Technology Department, University of Granada, 18071 Granada, Spain; vcardenas@ugr.es; 5Independent Researcher, Montreal, QC H8Y 1N6, Canada; niurkadupotey@gmail.com; 6Department of Hospital Pharmacy, UZ Brussel, Laarbeeklaan 101, 1090 Jette, Belgium

**Keywords:** medication review, medicines use review, medication therapy management, drug utilization review, community pharmacy services, pharmaceutical services, narrative review

## Abstract

Background: Medication review is a multifaceted service aimed at optimizing the use of medicines and enhancing the health outcomes of patients. Due to its complexity, it is crucial to clearly describe the service, its variants, and its components to avoid confusion and ensure a better understanding of medication review among healthcare providers. Aim: This study aims to bring clarity to the origins, definitions, abbreviations, and types of medication reviews, together with the primary criteria that delineate key features of this service. Method: A narrative review approach was employed to clarify the diverse terminology associated with “medication review” services. Relevant references were initially identified through searches on PubMed and Google Scholar, complementing the existing literature known to the authors. Results: The study uncovers a complicated and sometimes convoluted history of “medication review” in different regions around the world. The initial optimization of medicine use had an economic purpose before evolving subsequently into a more patient-oriented approach. A selection of abbreviations, definitions, and types were outlined to enhance the understanding of the service. Conclusions: The study underscores the urgent need for comprehensive information and standardization regarding the content and quality of the services, collectively referred to as “medication review”.

## 1. Introduction

Looking back in history, the pharmacy profession has experienced significant growth, change, and development and has expanded its scope of practice. Pharmacy was seen as a bridge between the health and chemical sciences. Historically, pharmacists crafted drug products *secundum artem* (according to the art), mostly for medicinal purposes [1]. By the 1950s, the pharmaceutical industry’s mass production and the enforcement of prescription-only legal status for many therapeutic agents had reoriented pharmacists’ roles, focusing on medicine dispensing. In 1960, the concept of ”clinical pharmacy” was mentioned for the first time [2]. Interventions aiming at optimizing “medication use” were initiated, but this endeavor was often a largely economic systemwide activity invisible to the individual patient [3]. In 1990, in their pivotal and highly cited paper [4], Hepler and Strand, alarmed by the high prevalence of drug-induced hospital admissions, expressed the need for further professional reorientation to ensure safe and effective drug therapy through pharmaceutical care as a new philosophy of patient-centered practice [4]. Pharmacy services became gradually more patient-centered, but pinpointing the exact origin and first mention of “medication review” in the literature is rather challenging. To address this challenge, it is necessary to clarify some confusion about word usage and definitions.

Searching “medication review” by country, using the search options in Google, produces a wide array of results that can easily lead to confusion among people unfamiliar with the topic. However, one of the searches yielded a valuable definition sourced from the guidelines of the National Institute for Health and Care Excellence (NICE): “Medication review is a structured, critical examination of a person’s medicines with the objective of reaching an agreement with the person about treatment, optimizing the impact of medicines, minimizing the number of medication-related problems and reducing waste” [5].

The complex nature of “medication review” is evident in its comprehensive scope, encompassing the identification of medication-related issues, exploring a patient’s pharmaceutical history, ensuring continuous data exchange among healthcare providers, incorporating the Social Determinants of Health (SDOH), and facilitating consultations between these providers and the patient [6,7,8]. All these efforts aim to optimize the patient’s use of medications in the face of the ever-expanding complexity of the pharmacotherapeutic landscape. It is, however, crucial to define the content, common language, and definitions of “medication review” before making comparisons, especially when prioritizing quality and examining stringent endpoints [9,10,11].

## 2. Materials and Methods

For this investigation, a narrative review approach was adopted to clarify the diverse terminology associated with “medication review” services and to offer a historical context for this pharmaceutical service within primary care. Initially, pertinent references were identified by searching PubMed and Google Scholar, supplementing the literature already known to the authors. The keywords used included “medication review”, “medicines use review”, “medication therapy management”, and “drug utilization review”. Subsequently, the search was broadened through citation tracking [12]. Finally, the grey literature was explored, mainly using Google Scholar and citation tracking, for various definitions and guidelines related to medication review and other pharmacy services, including drug utilization review.

Considering the substantial heterogeneity in terminology, procedures, contexts, and outcomes associated with the research question, the application of a traditional systematic meta-analysis was deemed unsuitable.

## 3. Results

### 3.1. History of Medication Review

By the mid-1960s, pharmacists transitioned towards a more patient-centered approach, introducing the concept of clinical pharmacy [1,2,4], the beginning of an evolution detailed further in Figure 1. Pharmacists actively participated in optimizing patient medication therapies within hospital settings [13]. While pharmacists have been examining medicine charts and offering recommendations to prescribers since the 1980s, this practice was not yet widely adopted in primary care settings during that period [4,14].

Nevertheless, there were early efforts in the 1960s in the United States to implement “drug utilization reviews” (DURs), laying the groundwork for the current concept of medication reviews (MRs) [14,15]. In 1969, the US governmental commission named the Task Force on Prescription Drugs published a document titled “Approaches to Drug Insurance Design: Background Papers”, marking an early significant milestone in the realm of DUR or MR [16]. To reduce confusion over terms and abbreviations, Table 1 presents an overview of many of these related services linked somehow to “medication review”.

DUR is an authorized, structured, ongoing review of prescribing, dispensing, and medicine use [17]. DUR and related procedures were concerned with monitoring and assessing population-level medication utilization [15] to ensure its quality and cost-effectiveness [3,14].

**Table 1 pharmacy-12-00039-t001:** List of used abbreviations and their respective services related to medication review.

Abbreviation	Service	Description
**AUR**	Antibiotic Utilization Review	DUR performed among hospitalized patients treated with antibiotics [18].
CMM CMTM	Comprehensive Medication Management Comprehensive Medication Therapy Management	An individualized care plan to achieve the intended goals of therapy with appropriate follow-up to determine actual patients’ outcomes, involving their active participation [19]. A CMM program includes several similar elements to a CMR, yet it extends its scope to address additional facets of the patient’s overall care [20]. CMM not only incorporates the patient’s history into recommendations, similar to CMR, but it also aims to influence elements of that history through measurable clinical outcomes [20].
CMR	Collaborative Medication Review	An internationally accepted term for medication review practices involving pharmacists collaborating closely with other healthcare professionals to review patients’ medicines. Their shared goal is to optimize the use of medications and prevent inappropriate medication use [21].
CMR	Comprehensive Medication Review	A comprehensive, annual, systematic review of all available patient-specific information and medication assessments to identify and resolve potential medication-related problems. CMR involves collaboration between the patient, pharmacist, and prescriber to determine appropriate options for resolving identified problems [22,23].
CMS	Chronic Medication Service	A service established at pharmacies in Scotland dedicated to helping patients with long-term conditions manage their medicines [24].
DRUM	Dispensing Review of Use of Medicines	A review of the use of medicines with the purpose of helping patients understand their medicines and identify medicine-related problems [6].
DUE	Drug Use Evaluation Drug Usage Evaluation	A group of structured reviews of prescribing, dispensing, and use of medication to ensure their appropriate and safe use while also optimizing the economic aspect of drug utilization [15,18].
DRR	Drug Regimen Review	
DUR	Drug Utilization Review Drug Use Review	
MUE	Medication Use Evaluation	
MUM	Medication Use Management	
DMMR HMR	Domiciliary Medication Management Review Home Medicines Review	An Australian MR program involving pharmacists conducting a domiciliary visit to review patients medications [25,26].
MAP	Medication-related Action Plan	One of the core elements of an MTM service; it is a patient-centered document equipped with a list of action steps for the patient to use in tracking progress for self-management of medication-related problems [23].
MR	Medication Review	A structured evaluation of a patient’s medicines with the aim of optimizing medicine use and improving health outcomes [27].
MR1	Medication Review type 1	
MR2	Medication Review type 2 *Intermediate medication review*	
MR3	Medication Review type 3 *Clinical medication review* *Advanced medication review*	
MRF	Medication Review with Follow-up	An ongoing and structured assessment of the patient’s pharmacotherapy performed in Spain that comprises detection of drug-related problems and negative outcomes related to medicines (NOMs), the development of a care plan, and monthly follow-up to provide continuing care [28].
MTA	Medicines Therapy Assessment	A clinical MR program conducted in New Zealand by pharmacists in collaboration with prescribers to review the use and understanding of prescribed therapy, identify medication-related problems, and work with the patient and wider healthcare team to resolve these issues and optimize medication use [29].
MTM MTMS	Medication Therapy Management Medication Therapy Management Services	A distinct service or group of services to optimize therapeutic outcomes for individual patients [30]. The MTM service model can be divided into the five core elements: Medication Therapy Review (MTR), intervention and referral, Personal Medication Record (PMR), Medication-related Action Plan (MAP), and documentation and follow-up [23].
MTR	Medication Therapy Review Medicine Therapy Review	One of the core elements of an MTM service; a systematic process that involves collecting patient-specific information, evaluating medication therapies to identify medication-related problems, creating a prioritized list of these problems, and devising a resolution plan. MTR can be comprehensive (CMR) or targeted (TMR) [23].
MUR	Medicines Use Review	A subtype of MR where pharmacists partner with patients to improve their medicine use and adherence [31]. Referring to the Pharmaceutical Care Network Europe (PCNE) definition, type MR2a includes MUR [32].
NMS	New Medicine Service	A service providing help and advice about medicines to patients who are prescribed a medicine to treat a long-term condition for the first time [33].
PMR	Personal Medication Record	One of the core elements of an MTM service; it contains an up-to-date list of medications, helping patients manage their pharmacotherapy [20].
QUM	Quality Use of Medicines	A package of services performed by Australian pharmacists to support the quality use of medicines, including HMR and RMMR [26].
RMMR	Residential Medication Management Review	An Australian program involving pharmacists conducting MRs of patients residing in aged care facilities [26,34].
SMR	Structured Medicine Review	A review of a patient’s medication, taking into consideration all aspects of the patient’s health in the form of shared decision-making conversations between a clinician and a patient [35].
TMR	Targeted Medication Review	Ongoing medication monitoring to assess medication use and identify and address specific actual or potential medication-related problems [20]. TMR involves follow-up with a healthcare professional or a patient to resolve identified medication-related problems. TMR must be performed quarterly, which enables identifying issues on a more regular basis than through yearly CMR [20].

In 1970, the first DUR program was conducted by a private pharmaceutical company in the United States [14]. Three years later, in 1973, the US-based organization, the Joint Commission (formerly known as JCAHO—the Joint Commission on Accreditation of Hospitals), introduced Drug Usage Evaluation (DUE), which was a more advanced analysis of medications, their uses, and their contributions to various patients’ outcomes. DUE represented an interdisciplinary and systematic approach to evaluating and improving medication use, particularly at the patient level, unlike DUR. Another term, Drug Regimen Review (DRR), one of the earliest examples of DUR, was introduced in 1974 as part of a quality assurance program for the care of Medicaid recipients in the United States. During the program, pharmacists were required to conduct monthly drug reviews among nursing home residents. They would assess patients’ prescribed medications to identify any potential drug-related problems, and then provide recommendations to the healthcare team for adjustments to the drug regimen [15].

In the 1980s, the Joint Commission made the implementation of Drug Use Evaluation evaluation (DUE) into hospital procedures one of the items to be audited. Initially, the program would only focus on the use of antibiotics for hospitalized patients. Over time, this evaluation process expanded to all medications [15,18]. Another concept that contributed to the development of MR was a “brown bag review”. The method was developed in 1982 under the name of the “Brown Bag Prescription Evaluation Program” in the United States [36]. Its name originates from the brown supermarket bags in which patients would bring all their medications and supplements that they had at home (including those prescribed by physicians, over-the-counter medications, supplements, or complementary medicines) to a healthcare appointment [37]. A healthcare professional, usually a pharmacist or physician, would review all patients’ medications comprehensively, address and resolve any medication-related problems, and, ultimately, educate the patient about the proper use of medications [36,37]. The “brown bag review” presented a new, patient-centered approach and closely mirrored some of the objectives of today’s MR.

Due to the Omnibus Budget Reconciliation Act of 1990 (OBRA ’90), pharmacists in the United States were mandated to incorporate DUR outside hospitals as part of their healthcare for Medicaid beneficiaries [18,38]. All the DUR-related programs served similar functions. Nevertheless, the concepts of DRR and DUE relied on reviewing the appropriateness of an individual patient’s therapy, whereas DUR constituted an analysis of a larger number of prescription profiles [15]. The term “drug utilization review” can easily be confused with “drug utilization research” or “drug utilization studies”, concepts embraced within the pharmacoepidemiology discipline that are time-limited investigations focused on measuring drug usage, without necessarily evaluating individual appropriateness or attempting to bring about changes in a particular patient’s therapy [14].

The transition to the 2000s represented a notable period of progress for MR programs [7], illustrative of the momentum of the pharmaceutical care movement that started a decade earlier [4]. Projects were initiated in Australia, the United Kingdom, Switzerland, New Zealand, the United States, Canada, the Netherlands, Germany, Sweden, and Denmark [7]. Moreover, the Beers Criteria for Potentially Inappropriate Medication Use in Older Adults, published in 1991 by Mark H. Beers, an American geriatrician, and updated by the American Geriatrics Society (AGS) in the following years, provided a standardized tool for identifying potentially inappropriate medications and improving medication management [39]. Since that time, numerous other tools have been developed to enhance the medication management of the elderly. A study conducted in 2019 identified a total of 76 such tools [40]. Among these additional tools, START/STOPP stands out as one of the most widely acknowledged, having been established in 2008. Since its inception, it has undergone two subsequent versions, and its implementation is often customized to align with the specific contexts of different countries [41]. In addition to these explicit tools, an implicit method is also used, which involves the Medication Appropriateness Index (MAI) to identify potential inappropriate prescriptions. Although this implicit method may be more time-intensive and challenging to implement, it has the potential to be more comprehensive [42,43].

In the early 1990s, national health departments and related entities, as well as various international organizations, such as the World Health Organization (WHO) and the International Pharmaceutical Federation (FIP), developed the first guidelines and frameworks to implement medication reviews in primary care [44]. All these initiatives throughout the years reinforced the role and responsibility of pharmacists in patient care and drug therapy management service delivery.

The first countries to formally integrate medication review into primary care were Australia (2001), the United States (2003), and the United Kingdom (2005) [26,45]. The Australian Home Medicines Review (HMR), also known as the Domiciliary Medication Management Review (DMMR), was launched in 2001, and it was perceived as a forerunner to many of the subsequent medication reviews [6,26]. The Australian pharmacists provided home visits to evaluate the patient’s current medication regimen and then consult with a clinician about any potential drug-related problems [46]. The Residential Medication Management Review (RMMR) was launched in Australia in 2005, providing medication reviews for occupants of these care facilities. The HMR and RMMR support the Quality Use of Medicines (QUM) initiative in Australia [26].

In 2003, the American Pharmacists Association (APhA) introduced the concept of Medication Therapy Management (MTM) in the United States [47] as a group of services to optimize therapeutic outcomes for individual patients [47]. Comprehensive medication review constitutes one of the pharmaceutical services within the MTM [48]. Other examples of MTM are intervention and referral, a personal medication record, a medication-related action plan, documentation, and follow-up [20]. In the UK, government policy documents, including the National Service Framework for Older People, have integrated medication reviews into primary care [49], with Medicines Use Reviews (MURs) introduced in England and Wales in 2005, and the Chronic Medication Service in Scotland in 2010 [6]. Another evaluation—the New Medicine Service (NMS)—was launched in 2011 to improve the adherence and outcomes of patients starting new medications [33]. Two years later, the National Health Service (NHS) published its overall “Medicines Optimisation Agenda” to improve patient outcomes through better use of medicines, with reviews of patients’ medication regimens as one way to reach that goal [50]. Nevertheless, despite the presence of shared features and objectives, these reviews exhibited notable variations in the terminology employed. In 2009, the Pharmaceutical Care Network Europe (PCNE) established a “medication review working group” to standardize the terminology and practice of this service performed by pharmacists [27]. Subsequently, a global spread of medication review projects can be recorded [51,52,53]. Starting in 2016, pharmacists have been able to retrieve a Summary Care Record (SCR) containing crucial clinical details, such as medication history, allergies, and adverse reactions, sourced from the patient’s GP record [54]. In April 2021, the Medicines Use Review (MUR) program was discontinued in the UK and replaced by Structured Medication Reviews (SMR) [55].

Considering more recent events, the COVID-19 pandemic has highlighted the community pharmacists’ proficiency in identifying and effectively addressing medication-related problems [56] and ensuring the safe and effective use of long-term medications [56,57]. Together with the pandemic’s demand for testing and vaccinations, this has further underscored the indispensable non-dispensing-related roles of community pharmacists.

### 3.2. Definitions and Various Types of Medication Review

As the implementation of medication reviews continues to grow, it is essential to clearly define what this pharmaceutical care service entails. The most important definitions are shown in Table 2. Zermansky et al. [58] in 2002 formulated one of the first definitions: “the process where a health professional reviews the patient, the illness, and the drug treatment during a consultation. It involves evaluating the therapeutic efficacy of each drug and the progress of the conditions being treated. Other issues, such as compliance, actual and potential adverse effects, interactions, and the patient’s understanding of the condition and its treatment, are considered when appropriate. The outcome of the review will be a decision about the continuation (or otherwise) of the treatment”.

The authors of the “Oxford Handbook of Clinical Pharmacy” of 2007 also presented a concise and useful definition: “a structured critical examination of a patient’s medicines by a healthcare professional reaching an agreement with the patient about treatment, optimizing use of medicines, minimizing the number of drug-related problems, and avoiding wastage” [59].

**Table 2 pharmacy-12-00039-t002:** The most important definitions of medication review, sorted by publication date.

Definition	Source	Year
A structured, critical examination of a patient’s medicines with the objective of reaching an agreement with the patient about treatment, optimizing the impact of medicines, minimizing the number of medication-related problems, and reducing waste [60].	Medicines Partnership	2002
A structured, critical examination of a patient’s medicines by a healthcare professional: reaching an agreement with the patient about treatment, optimizing the use of medicines, minimizing the number of medication-related problems, avoiding wastage. Regular medication review maximizes the therapeutic benefit and minimizes the potential harm of drugs. It ensures the safe and effective use of medicines by patients. A medication review provides an opportunity for patients to discuss their medicines with a healthcare professional. Medication review is the cornerstone of medicine management [59].	Oxford Handbook of Clinical Pharmacy, 1st edition	2007
A structured, critical examination of a person’s medicines with the objective of reaching an agreement with the person about treatment, optimizing the impact of medicines, minimizing the number of medication-related problems, and reducing waste [31].	National Prescribing Centre (NPC)	2008
A structured evaluation of a patient’s medicines with the aim of optimizing medicine use and improving health outcomes. This entails detecting drug-related problems and recommending interventions [27].	Pharmaceutical Care Network Europe (PCNE)	2018

Nowadays, an often-used definition is the one developed by PCNE in 2018 that characterizes medication review as a structured evaluation of a patient’s medicines with the aim of optimizing medicine use and improving health outcomes. This definition entails detecting drug-related problems and recommending interventions [27].

The initial classification of various levels of medication review that received significant recognition was introduced in 2002 in “Room for Review”, published by the National Prescribing Centre (NPC), an NHS organization supported by the British Department of Health [31,60]:Level 0—Ad hoc—unstructured, opportunistic review.Level 1—Prescription review—a technical review of the list of a patient’s medicine;Level 2—Treatment review—a review of medicines with the patient’s full notes;Level 3—Clinical medication review—a face-to-face review of medicines and conditions.

Subsequently, the classification that has gained widespread acceptance was published in 2008 within the NPC’s updated document, “A Guide to Medication Review”. In accordance with this classification, the following types of MR were delineated [31]:Type I—Prescription review—addresses technical issues relating to the prescription; the patient is usually not involved; it is a review of medicines.Type II—Compliance and concordance review—addresses issues relating to the patient’s medicine-taking behaviors; the patient is usually involved; it focuses on medicine use. This type includes MURs.Type III—Clinical medication review—addresses issues relating to the patient’s use of medicines in the context of their clinical conditions; the patient is always involved, and there is also always access to patient information (e.g., clinical conditions and laboratory test results). It reviews medicines and conditions.

In some countries, the extension of Type III—Clinical review with prescribing, also known as Type IV, exists as well and includes prescribing authority [53].

Currently, the classification published by PCNE in 2018, which divided medication reviews into three types, is in widespread use [32]:Type 1—Simple MR (MR1)—is based solely on the patient’s medication history available in the pharmacy; it enables the detection of drug interactions, some side effects, unusual dosages, and some adherence issues. This type of MR is part of routine dispensing.Type 2—Intermediate MR—classified into two subtypes: -Type 2A (MR2A)—based on the medication history and patient information; thus, it is useful when the patient can be interviewed; it detects drug interactions, drug–food interactions, side effects, unusual dosages, effectiveness, and adherence issues, but also issues with OTC medications.-Type 2B (MR2B)—based on the medication history and clinical information obtained from the general practitioner (GP) or physician; detects drug interactions, drug–food interactions, side effects, unusual dosages, adherence issues, effectiveness issues, indication without a drug, and drugs without indication.Type 3—Advanced or Clinical MR (MR3)—based on a complete medication history, an extensive patient interview, and clinical data obtained from the GP or the physician; detects drug–drug interactions, drug-food interactions, issues with OTC drugs, side effects, unusual dosages, adherence issues, effectiveness issues, indication without a drug, and drugs without indication.

Nevertheless, the definition, comprehensiveness, levels of interprofessional collaboration, and remuneration of MR still vary among different countries, mainly due to their specific processes, guidelines, and terminology [10].

Table 3 presents an overview of several key guidance documents about MR. MR may also be associated with other pharmaceutical or medication management services, such as medical reconciliation, deprescribing interventions, or the previously mentioned drug utilization reviews.

### 3.3. The Principal Criteria Delineating Key Features of Medication Review

#### 3.3.1. Participating Healthcare Providers

A MR should be carried out by a skilled healthcare professional. Current literature and practice suggest that pharmacists, GPs, or nurses are typically the ones conducting MRs, listed in descending order of prevalence [72].

Interprofessional collaboration is a fundamental component of the MR process and enhances its quality by providing a more comprehensive understanding of patients and their medications [6,73,74]. Healthcare providers must trust the reviewing practitioner and engage in open discussions about potential recommendations to prescribers during patient regimen evaluation. Relying solely on written recommendations from pharmacists to GPs is less effective, highlighting the importance of a strong collaboration between GPs and community pharmacists for effective MRs [6,75].

Another significant factor pertains to the patients’ accessibility to community pharmacists [76,77]. Diversity in healthcare structures among various countries may contribute to variations in how MRs are carried out [51]. For instance, in some countries, patients can closely cooperate with community pharmacists, who monitor their pharmacotherapy and guide them in medication use [20,26,75].

#### 3.3.2. Target Group of Patients

Currently, there are no globally recognized standards yet that conclusively identify the patients who are to be prioritized for MRs. Eligibility can differ based on the country and healthcare system, and it usually depends on a combination of factors that have been correlated with drug-related problems such as multimorbidity, the complexity of the medication schedule (including polypharmacy), the patients’ age and frailty, and the presence of high-risk medicines [78,79]. Effectiveness research should ultimately determine who benefits most. Furthermore, it is essential to recognize that compensation should be proportionate to the complexities of the case, incorporating the social determinants of health. There is a rising recognition that healthcare should consider individuals’ physical, mental, and socioeconomic well-being, taking into account subjective experiences, and recognizing the SDOH to effectively address drug-related problems [80]. The integration of this additional dimension substantially increases the complexity of MR. Medicines are for real people who grapple with real-world problems, leading to less-than-ideal adherence and an array of preventable drug-related problems. Failing to incorporate this into a patient-centered pharmaceutical care philosophy will result in an inadequately powered MR and a significant number of patients being denied the full benefit of pharmacotherapy.

#### 3.3.3. The Most Crucial Outcome Studies

The process of MR offers a diverse range of potential advantages, including clinical, economic, humanistic, and other related outcomes. Although the purported effects of MRs appear realistic and achievable, irrefutable proof from RCTs substantiating their positive outcomes remains scarce and, in some cases, inconclusive. Among the positive effects of MRs, the most consistent and substantial are the reduction of inappropriate prescriptions, the reduction of drug-related problems, and increased adherence [81,82,83].

The effects on mortality and morbidity related to MR have been studied, but there is a lack of unequivocal findings [83]. In the case of hospitalizations, the outcomes are inconsistent [81,83]. Nevertheless, Mizokami et al. [84] reported that MR interventions might be effective in inpatient settings but found no such results in outpatient settings [84]. Moreover, the same authors suggested that a reduction in hospital admissions was more likely for MR3 as compared to MR1 and MR2 [84]. There are a small number of studies presenting a positive impact on the level of laboratory values such as low-density lipoproteins, cholesterol, and HbA1c, as well as blood pressure [52,53,85]. Moreover, MR contributed to the decrease in the number of falls among patients [82]. However, studies on patients with frailty have not provided clear conclusions yet [86], but as there is no universally accepted definition or assessment of frailty, this is not very surprising [87].

In terms of the patient’s quality of life, most studies showed no significant impact on this aspect, apart from one systematic review that reported the benefit of MTM services on patients’ physical outcomes, while minimal effect was observed in mental outcomes [81,82,83].

Regarding the impact of MR on cost-effectiveness, the evidence is also limited and mixed. A small number of studies have demonstrated substantial cost savings due to reduced healthcare utilization and medication expenditures [52,81,83].

## 4. Discussion

The history of MR reflects a distinct and extensive journey to reach its current state. In its early stages, approaches to drug optimization, such as DURs, were largely driven by economic considerations. In contrast, MRs are now primarily focused on improving and ensuring the optimization of patient treatment, complemented by patient education initiatives [3].

MR extends beyond a mere definition. Although the content of MRs can vary widely, different types often converge on similar principles, as shown in previous studies [10,11]. For example, Medication Therapy Management (MTM) comprises various components, with MR being one of them [20]. Upon closer examination, many of the other components are frequently integral to MR. Conversely, the MR process in Spain, referred to as medication review with follow-up (MRF), involves not only a structured assessment of the patient’s pharmacotherapy but also ongoing monthly follow-ups [28]. The complex nature of MRs renders the assessment and comparison of tangible outcomes challenging, particularly in the absence of a standardized methodology and given the variations in processes and healthcare systems across projects and countries.

The lack of definitive evidence regarding the positive outcomes of medication reviews from randomized controlled trials has raised questions about its validity and necessity. First, it is worth noting that assessing the effectiveness of MRs is complex and poses challenges in study design and implementation. The process or feasibility of performing such studies is not immediately clear or simple. However, while randomized controlled trials (RCTs) are considered the gold standard for evaluating the effectiveness of a single well defined and easily replicable intervention, today these characteristics do not align with MRs. The complex and multifaceted nature of MR makes it challenging to measure or assess using RCTs. There remain, however, enough arguments from clinical expertise, guidelines, qualitative studies, observational studies, and simple logic to support and further invest the effectiveness of MRs [88,89].

Other complex interventions, like deprescribing, also encounter challenges in evaluating their effectiveness [90,91]. What MR and deprescribing mutually share is also the goal of combating inappropriate prescribing, which is the primary contributor to multimorbidity. It is not yet possible to compare or compile the results of trials assessing the effect of these services in a convincing meta-analysis. Many studies, for instance, limited the scope to specific outcome measures, selected different group of patients, included various times of follow-up. Furthermore, there is a vast discrepancy in MR terminology, not only when it comes to defining the process itself, but also regarding terms used to describe and define activities undertaken for this purpose. A standardization of the terms related to MR would enable researchers to compare data from similar interventions and studies [10].

Additionally, it is also crucial to ensure a comprehensive quality assessment of MRs [9] before launching into a large-scale reliable and repeatable evaluation of their outcomes. As demonstrated in this review paper, certain countries have a longer history of conducting various forms of medication reviews, while others are just embarking on this journey [7,51]. Standardization holds the potential to improve the reliable implementation of this practice in more countries, similar to the benefits observed in adherence research through the definition of adherence terms [92].

## 5. Conclusions

This review paper described the origins, variety, types, and historical background of “medication review”. Additionally, it aimed to enhance its comprehension by collecting definitions and compiling a list of guidelines about the MR process. Although Blenkinsopp et al. [6] reported a decade ago on the state of the art of MR in the UK, this review paper explored MRs from an international perspective while pointing towards the progress made in recent times.

Regardless, there remains a pressing need for internationally supported standardization and a more comprehensive description of the service’s content and quality to enable comparisons between studies and facilitate a broader implementation. This should also allow more reliable assessments of MRs’ outcomes and strengthen the uptake of this service, all with the final goal of improving pharmaceutical care for patients with complex medication needs.

## Figures and Tables

**Figure 1 pharmacy-12-00039-f001:**
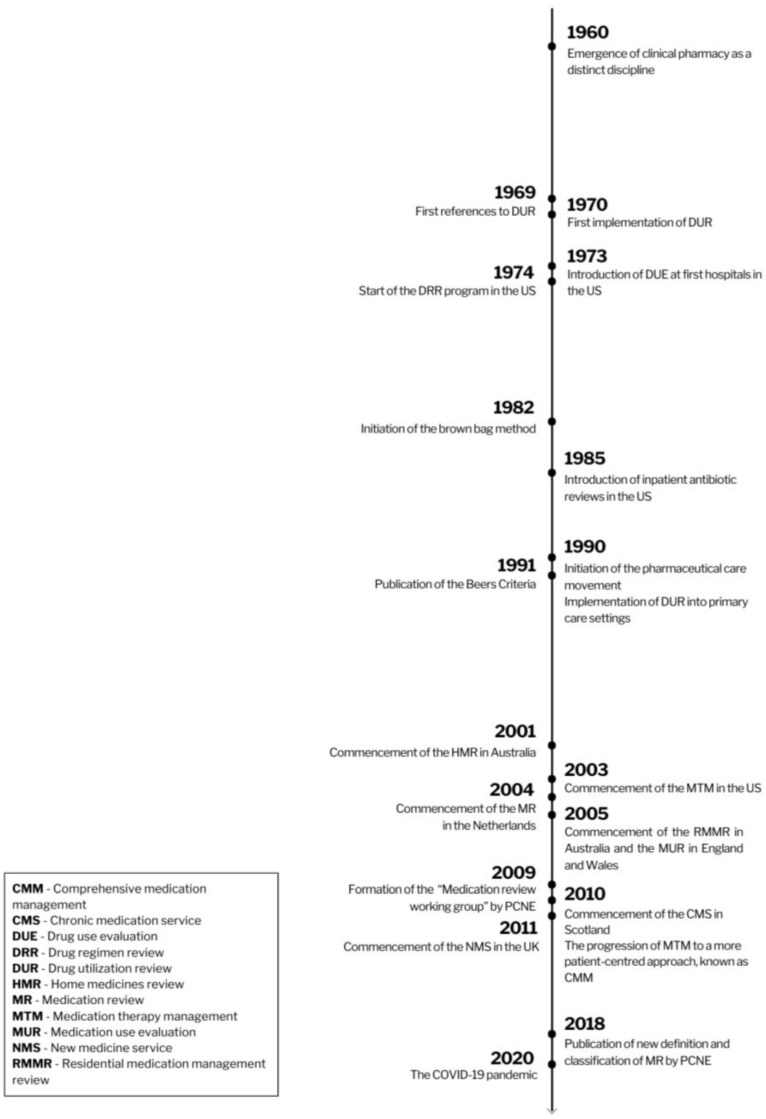
The history of medication review.

**Table 3 pharmacy-12-00039-t003:** List of various guidance documents concerning medication review.

Organization and Country	Guideline	MR Type by PCNE	Year
American Pharmacists Association; National Association of Chain Drug Stores Foundation, USA	Medication therapy management in pharmacy practice: core elements of an MTM service model (version 2.0) [23]	3	2008
Patient-Centered Primary Care Collaborative, USA	The Patient-Centered Medical Home: Integrating Comprehensive Medication Management to Optimize Patient Outcomes Resource Guide [19]	3	2012
Saskatchewan Ministry of Health, Canada	Saskatchewan Medication Assessment Program (SMAP). Procedures and guidelines for Saskatchewan pharmacists [61]	3	2013
Royal Pharmaceutical Society, UK	Medicines Optimization: Helping patients to make the most of medicines [50]	2a	2013
National Institute for Health and Care Excellence (NICE), UK	Medicines Optimization: The Safe and Effective Use of Medicines to Enable the Best Possible Outcomes [5]	2a	2015
Ontario Ministry of Health and Long-Term Care, Canada	Professional Pharmacy Services Guidebook 3.0. MedsCheck, Pharmaceutical Opinion and Pharmacy Smoking Cessation Program [62]	2a	2016
Comprehensive Medication Management in Primary Care Research Team, USA	The Patient Care Process for Delivering Comprehensive Medication Management (CMM): Optimizing Medication Use in Patient-Centered, Team-Based Care Settings [63]	3	2018
Pharmaceutical Society of Ireland (PSI), Ireland	Guidelines on the Counselling and Medicine Therapy Review in the Supply of Prescribed Medicinal Products from a Retail Pharmacy Business [64]	2a	2019
Pharmaceutical Society of Australia (PSA), Australia	Guidelines for pharmacists providing Residential Medication Management Review (RMMR) and Quality Use of Medicines (QUM) services [65]	3	2019
Pharmaceutical Society of Australia (PSA), Australia	Guidelines for Quality Use of Medicines (QUM) services [66]	3	2020
Pharmaceutical Society of Australia (PSA), Australia	Guidelines for comprehensive medication management review [67]	3	2020
The Royal Dutch Pharmacists Association (KNMP), The Netherlands	Guideline for conducting clinical medication review in community pharmacy [68]	3	2020
National Health Service (NHS), UK	Structured medication reviews and medicine optimization: guidance [69]	3	2020
Department of Health and Aged Care, Australia	Guiding Principles for Medication Management in the Community [70]	3	2022
General Pharmaceutical Council of Spain, Spain	Practical guide to Clinical Professional Pharmacy Services (CPPS) in Community Pharmacy [71]	3	2022

## Data Availability

Data sharing is not applicable to this article.

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
