# Peer review of "Medication Review: What’s in a Name and What Is It about?"

_pharmacy, 2024, doi:10.3390/pharmacy12010039_

Round 1

Reviewer 1 Report

Comments and Suggestions for Authors

Thank you for an interesting paper on the variations of the term 'medication review'. The variation in the naming and content of the reviews was remarkable. I only have a few fairly minor suggestions:

Page 2, lines 52/53 starting 'One of the searches...' Please correct the sentence as it didn't make sense.

Results - when talking about the history of the medication review, please can you include the country the particular review was introduced in. For example, you mention in the 1960's there was the DUR, but have not stated where. The same goes for other parts of this section - JCAHO, Beers Criteria etc..

Table 1 - MUR: please define PCNE

Page 7, line 179: the MUR has now been removed as a service. The NMS is still around.

Something to potentially include is that within England, community pharmacies can have access to summary care records, which allows them to be better informed when undertaking a medication review. https://cpe.org.uk/digital-and-technology/electronic-health-records/summary-care-record-scr/

Page 11, line 299 - please correct the first sentence 'There aren't no....'

Page 12, line 377 - do you have any suggestions for a standardisation of terms related to MR based on what you've found?

Page 8, lines 223: NPC is from where?

Author Response

We thank the reviewer for his appreciation of our manuscript. We are also indebted to the reviewer for his careful reading and the insightful comments, which allowed us to improve the quality of our paper. We acknowledge the majority of the comments and changed the manuscript accordingly.

Reviewer 2 Report

Comments and Suggestions for Authors

This manuscript presents a historical timeline of the use of the terminology related to medication reviews.  It was an interesting read, especially the history of medication reviews and the changes in terminology as well as the components of the medication review. I have a few minor questions and suggestions for the authors:

1.     Under section 2: Materials and Methods: the authors should further describe their search strategy; for example, how many pages of google did they search, how did they determine which articles to include and exclude, how did they conduct grey literature search (was this outside the google search? If so, which databases did they search?

2.     While the authors mention START and STOP, they do not delve into other implicit medication review instruments such as the Medication Appropriateness Index (MAI), IPET, etc.  It may be useful to provide some background and historical context and place in conducting medication reviews for these instruments as well.

3.     Results related to the impact of medication reviews appear to be synthesis of studies – did the authors use a systematic methodology to synthesize their findings?

Comments on the Quality of English Language

English grammar and syntax errors all dispersed through the manuscript and would benefit from a thorough review and editing. For example, do the authors mean “incorrect prescribing” or inappropriate prescribing?  What do the authors mean by “exemplified in this review”?  There are other examples of such language where it is not clear how to interpret the sentence/paragraph.

Author Response

(The authors gave the same response as above.)
